# Image-based high-throughput mapping of TGF-β-induced phosphocomplexes at a single-cell level

Peter Lönn [1,3 ✉], Rasel A. Al-Amin [1,3], Ehsan Manouchehri Doulabi [1], Johan Heldin [1,2], Radiosa Gallini[1], Johan Björkesten[1], Johan Oelrich[1], Masood Kamali-Moghaddam [1] & Ulf Landegren [1 ✉]

Protein interactions and posttranslational modifications orchestrate cellular responses to e.g. cytokines and drugs, but it has been difficult to monitor these dynamic events in high-throughput. Here, we describe a semi-automated system for large-scale in situ proximity ligation assays (isPLA), combining isPLA in microtiter wells with automated microscopy and computer-based image analysis. Phosphorylations and interactions are digitally recorded along with subcellular morphological features. We investigated TGF-β-responsive Smad2 linker phosphorylations and complex formations over time and across millions of individual cells, and we relate these events to cell cycle progression and local cell crowding via measurements of DNA content and nuclear size of individual cells, and of their relative positions. We illustrate the suitability of this protocol to screen for drug effects using phosphatase inhibitors. Our approach expands the scope for image-based single cell analyses by combining observations of protein interactions and modifications with morphological details of individual cells at high throughput.

[1] Department of Immunology, Genetics and Pathology, Science for Life Laboratory, Uppsala University, Uppsala, Sweden. [2] Department of Pharmaceutical Biosciences, Uppsala University, Uppsala, Sweden. [3]These authors contributed equally: Peter Lönn, Rasel A. Al-Amin. ✉email: peter.loenn@gmail.com; ulf.landegren@igp.uu.se

Cellular responses to stimulation or inhibition are reflected in the dynamics of protein interactions and posttranslational modifications (PTMs)[1–4]. Collectively, these dynamic events control signaling cascades and gene expression patterns. Monitoring and detailing such actions can reveal pathway-specific effects of disease or responses to targeted therapies[1,5]. Accordingly, unbiased and scalable means to record these dynamic events across many individual cells in large numbers of samples are needed in research, drug development, and ultimately also in clinical routine.

The in situ proximity ligation assay (isPLA) serves to visualize proteins, their PTMs, or proximity between two or more proteins over distances estimated at 40 nm or less[2,6,7]. The method has been extensively applied to validate interactions and modifications of endogenous proteins in fixed cells and tissues, but typically only for analyzing small numbers of conditions at a time. Recently, automated microscopy was used to study interactions between 60 proteins and the nuclear lamina[8].

Here, we use isPLA, automated microscopy, and computer-assisted image analysis to characterize and relate phospho-complex signaling with morphological features in millions of individual cells. We exemplify our approach by investigating the effects of transforming growth factor-β (TGF-β) on cultured cells. The TGF-β ligand binds and activates cell surface type I and type II TGF-β receptors, which in turn phosphorylate cytoplasmic Smad2 and Smad3 proteins at their C-termini, causing them to accumulate in the nucleus where they interact with Smad4[9,10]. Through further modifications and interactions with both proteins and nuclear DNA, the Smad complexes ultimately control the expression of gene sets. The signaling is finally terminated via protein relocation, dephosphorylation, and/or proteasomal degradation of the transcriptional complexes.

The TGF-β pathway controls several important cellular functions during embryonic development and in the adult organism, including cellular proliferation, differentiation, migration, and apoptosis[9,10]. TGF-β is commonly dysregulated in tumors, such as cancers of the breast, colon, and pancreas, thereby losing its normal growth suppressing function and instead promoting cancer progression[11–13]. Specific sites in the linker region of Smad2 and Smad3 are differentially phosphorylated to fine-tune transcriptional activity, control turnover, and regulate cross-talk with other signaling pathways[14–18]. Multiple kinases have been shown to act on these sites, including CDKs, GSK3β, and MAPKs[16–20]. The levels of Smad linker phosphorylations have also been seen to be affected by the cell cycle[19,21]. In vivo, Smad linker phosphorylation patterns have been connected with TGF-β-mediated invasion and metastasis[14,17]. Nonetheless, it remains unclear exactly how linker phosphorylations tune signaling by TGF-β. All these characteristics make linker-phosphorylated Smads important targets for single-cell analysis.

In summary, we present a semi-automated isPLA protocol to detail the life-cycle and dynamics of endogenous linker-phosphorylated Smad2 protein molecules across millions of individual cells, and we exemplify how this high-throughput platform can be used to screen for effects by targeted drugs.

## Results

### Visualizing temporal dynamics of phospho-Smad complexes using semi-automated isPLA.
We set out to establish a semi-automated isPLA protocol to investigate cellular signaling events at the single-cell level in high-throughput (Fig. 1a). For this purpose, we grew and fixed HaCAT keratinocyte cells in glass-bottom 96-well plates. The isPLA analysis uses pairwise antibody binding to trigger rolling-circle amplification, serving to generate rolling-circle products (RCPs) that reflect sites of proximal binding of the pairs of antibodies. The RCPs were visualized as bright fluorescent spots, recorded by automated microscopy, and enumerated using Cell-Profiler software[22]. To examine linker-phosphorylated Smad2 by isPLA, we combined antibodies recognizing Smad2 phosphorylated at T220 with either general anti-R-Smad antibodies (recognizing both Smad2 and Smad3) or with Smad4 antibodies. This allowed us to visualize either the total pool of pT220-phosphorylated Smad2 or those pT220-phosphorylated Smad2 proteins that colocalize with Smad4 (referred to as Smad2(pT220)/Smad4). The 96-well semi-automated format allowed us to monitor the TGF-β-induced increase and later decrease of Smad2(pT220) and Smad2(pT220)/Smad4 across individual cells (Fig. 1 b–f). Cells were starved and then treated variable times with TGF-β in triplicate wells before fixation and isPLA (Fig. 1b–d). Numbers of isPLA signals representing Smad2(pT220) increased from background levels by 20 min, peaked at 60 min, and slowly declined to about half maximum at 5 h (Fig. 1b, c, red line). Smad2(pT220)/Smad4 protein clusters also increased after 20 min, but peaked at 2 h, before declining to half maximum at ~5 h (Fig. 1b–d). Throughout the course of stimulation >70% of the signals overlapped with the nuclear staining, indicating a high degree of nuclear localization of the protein complexes (Fig. 1c, d, yellow line). The colocalization of RCPs and nuclear stains in the 2D images cannot exclude, however, that some of the RCP signals may also be located above or below the nuclei.

We next investigated the three serine residues at positions 245, 250, and 255, close to the T220 phosphorylation site on Smad2, which are also phosphorylated during the course of TGF-β signaling[18]. We designed isPLA tests of levels of Smad2(pS245/pS250/pS255) alone or in complex with Smad4 by combining specific anti-Smad2(pS245/pS250/pS255) antibodies with the same anti-R-Smad or anti-Smad4 antibodies described above. RCPs of both combinations increased after TGF-β stimulation (Supplemental Fig. 1a, b, red line), although the numbers of products were quite modest. This was especially true for the Smad2(pS245/pS250/pS255)/Smad4 complexes that were barely detectable above the background (Supplemental Fig. 1a, b).

As linker-phosphorylated Smads are known to undergo ubiquitination and proteasomal degradation[15,16,18,23], we hypothesized that rapid degradation of these phosphorylated proteins and complexes could contribute to the low signals we observed. Indeed, treating cells with the proteasomal inhibitor MG132 increased levels of phosphorylated proteins substantially (Fig. 1e, f and Supplemental Fig. 1c, d). This indicates that the phosphorylated complexes are rapidly processed by proteasomal degradation, rendering them short-lived and highly dynamic.

### Characterizing linker-phosphorylated Smad2 signaling at single-cell resolution.
The semi-automated high-throughput isPLA protocol enables large-scale single-cell analysis by relating the distribution of RCPs to other recorded properties of the individual cells (Fig. 1a). From each 96-well plate, we could collect up to 2592 images (nine fields of view/well × 96 wells × 3 wavelengths) at ×20 magnification, with typically ~300–1000 cells in each image. With multiple plates, this allowed us to process millions of individual cells in a relatively short time. Besides counting numbers of RCPs, which represent phosphorylated proteins and protein complexes for each cell, we also used the CellProfiler software to record cellular parameters in five categories (area, shape, intensity, texture, and neighbors), derived by monitoring the nuclear DNA staining. We quantified RCPs in relation to e.g. DNA content per cell, nuclear size and form, and the crowding of each cell.

We first visualized the range of RCPs for Smad2(pT220) and Smad2(pT220)/Smad4 per individual cell according to whether they had been stimulated with TGF-β or not (Fig. 2a and b). The

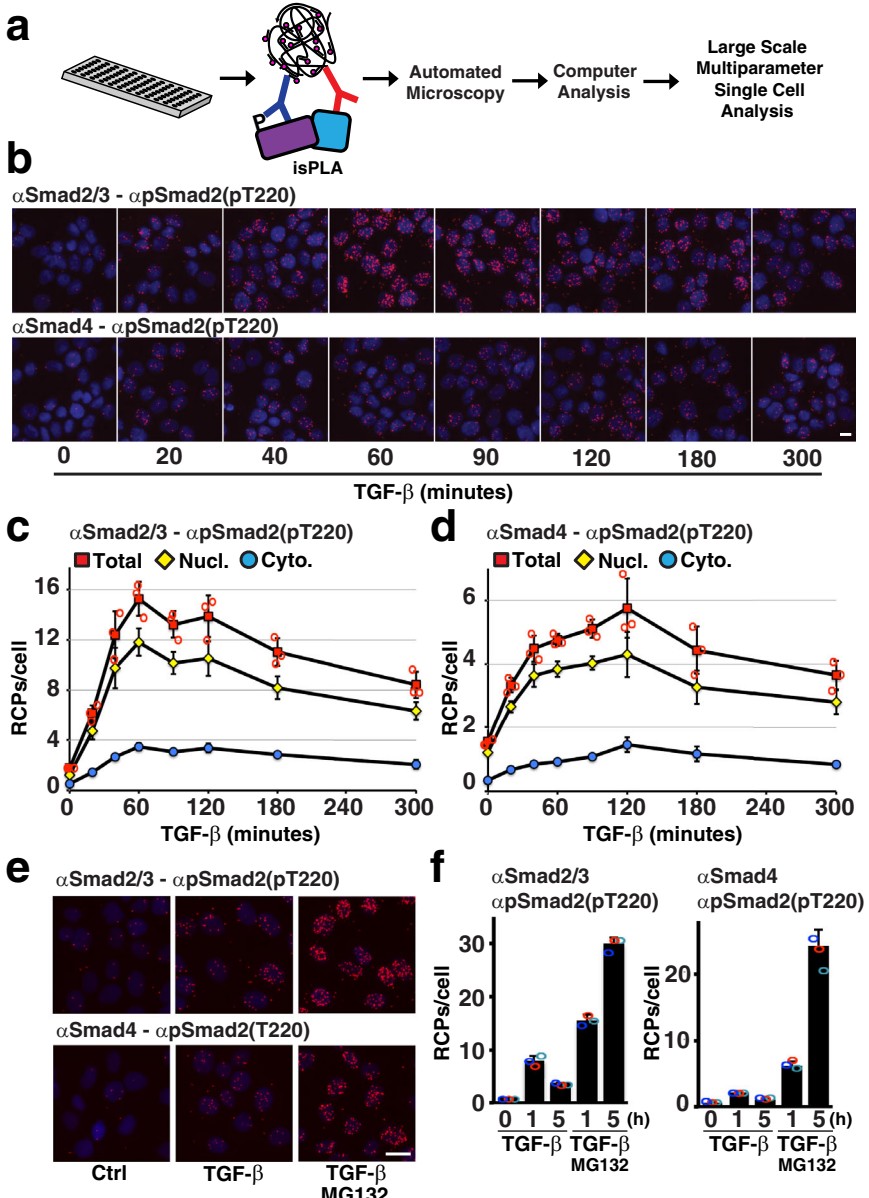

**Fig. 1 Smad2 pT220 linker phosphorylation was investigated using high-throughput isPLA. a** Overview of the semi-automated isPLA pipeline. **b–d** Time courses for the appearance of Smad2/3-Smad2(pT220) and Smad4-Smad2(pT220) reaction products upon stimulation with TGF-β. HaCAT cells were starved and treated with 10 ng/ml TGF-β for the indicated times before fixation, isPLA, and imaging. Results are displayed as representative fluorescence microscopy images with RCPs shown in red and nuclear DNA stained in blue (**b**), and as graphs showing mean RCP numbers per cell from cells grown in triplicate wells ($n = 3$), with nine images acquired and analyzed per well, and with standard deviations (SD) of the three analyzed wells displayed (**c**, **d**). The red squares indicate total average RCPs/cell, yellow diamonds indicate numbers of RCPs/cell that overlap with the nuclear stain, whereas blue circles represent RCPs/cells that do not overlap with the nuclear stain. Independent data points of the total average RCPs/cell for each well are also displayed as red circles in the graphs. **e** Fluorescence microscopy images of HaCAT cells starved and treated or not treated with 10 ng/ml TGF-β for 5 h with or without the proteasomal inhibitor MG132. **f** HaCAT cells were starved and treated with 10 ng/ml TGF-β for the indicated times with or without the proteasomal inhibitor MG132. Bar graphs show average RCPs/cell from cells grown in triplicate wells ($n = 3$), with nine images acquired and analyzed per well, and with SD of the three analyzed wells displayed. The independent data points of the total average RCPs/cell for each well are also displayed as red, blue, and turquoise circles in the graphs. Scale bars in **b** and **e** represent 10 μm.

numbers of RCPs for each cell were categorized along the $X$ axes and the total numbers of cells within each category were then visualized with bar graphs along the $Y$ axes (Fig. 2a and b). Stimulation with TGF-β led to a robust increase in cells with higher RCPs per cell. We also noted that the numbers of RCPs per cell in each condition vary over relatively wide ranges, likely reflecting contributions from both technical and biological variation. Although cell lines are fairly homogenous, individual cells nonetheless differ in factors that may influence the

investigated protein interactions and modifications, including cell cycle phase, degree of chromatin condensation, sizes of the cytoplasm and nuclei, and local cell density. The proteomic heterogeneity of the cell cycle phases has recently been systematically investigated and described[24].

To further investigate differences among the cells, we plotted individual cells according to their recorded nuclear sizes along the $X$ axis and the integral nuclear DNA intensity along the $Y$ axis, and we color-coded each cell according to its numbers of RCPs.

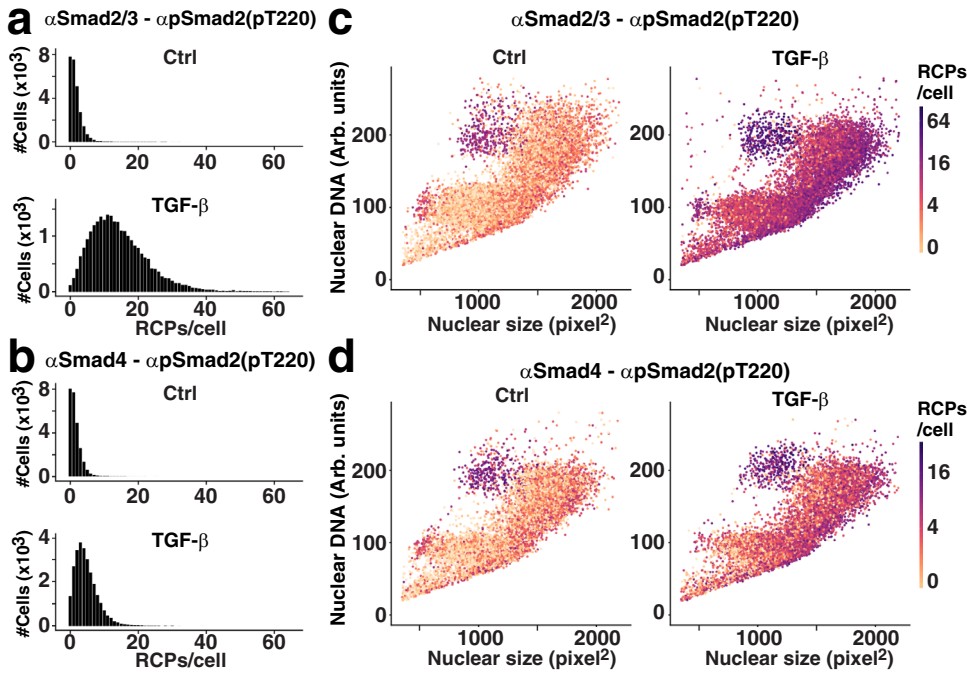

**Fig. 2 Smad2 linker phosphorylation and complex formation in individual HaCAT cells, starved and untreated or treated with 10 ng/ml TGF-β for 60 min. a, b** Results displayed as bar graphs according to the numbers of Smad2/3-Smad2(pT220) (**a**) or Smad4-Smad2(pT220) (**b**) RCPs/cell as categories along the X axes and with the height of the bars indicating the number of cells in each category according to the scale along the Y axes. The X axes were limited to display up to 65 RCPs/cell. **c, d** HaCaT cells, stimulated or not, were distributed in the plots according to the nuclear area of each cell along the X axes (pixels$^2$ (~0.11 μm$^2$)), and the integral intensity of the nuclear DNA stain along the Y axes. Numbers of Smad2/3-Smad2(pT220) (**c**) or Smad4-Smad2(pT220) (**d**) isPLA RCPs for individual cells were indicated by color scales. The log2 color scales indicate 0–96 RCPs/cell for **c** and 0–32 RCP/cell for **d**.

The plotted data displayed that the HaCAT cells have a quite regular pattern of nuclear shapes with one subcluster located around the arbitrary Hoechst intensity number 100 and with another subcluster situated around the 200-mark on the y axis. This pattern reflects that the genomes of the majority of cells are either 2 N or after genome duplication by replication 4N[25] as reflected both in the intensity of staining with DNA intercalating dyes, such as DAPI or as herein Hoechst, and in their nuclear sizes[26]. The cells between these clusters are likely replicating cells in S phase. This variation of nuclear staining by the Hoechst dye is similar to that commonly observed using either DAPI or Hoechst in the flow-cytometric analysis of the cell cycle. The color-coding of RCPs revealed a signaling pattern within the plots, where cells with larger nuclei consistently exhibited the greatest numbers of RCPs per cell (Fig. 2c, d (0 h and 1 h) and Supplemental Fig. 2). This was true across all time points. Furthermore, we also observed a subset of cells with smaller nuclei but high DNA content enriched for cells in mitosis. These cells were characterized by relatively high levels of Smad2 phosphorylations in both stimulated and unstimulated cells (Fig. 2c, d (0 h and 1 h) and Supplemental Fig. 2 (all time points)).

As DNA content per cells increases from 2 N to 4 N in preparation for mitosis[25] and nuclear sizes undergo a continuous increase from G1 to the G2 phase of the cell cycle[26], these two parameters together serve to roughly identify the location of individual cells within the cell cycle (Fig. 2c, d (0 h and 1 h) and Supplemental Fig. 2). The counter-clockwise pattern seen among the individual cells starts with newly divided 2 N cells with smaller nuclei in early G1 (lower left part of the graphs). The cells subsequently progress to larger 2 N nuclei in late G1 (lower right part of the graph). In the transition to S phase and G2 the DNA content increase reaching 4 N (middle right to the upper right part of the graph). This is followed by nuclear DNA condensation and mitosis (upper left part of the graph). Representative images of cell nuclei in the different areas of the graphs are shown in Supplemental Fig. 3, confirming the positioning of the nuclear morphologies described above. Interestingly, the images show that while RCPs are clearly overlapping with the nuclear stain during interphase, the RCPs are enriched around the metaphase plate in cells in mitosis, when the nuclear membrane collapses ahead of cell division, which has previously also been demonstrated for the related T179 linker phospho-site of Smad3[21] (Supplemental Fig. 3). The responsiveness of cells to TGF-β stimulation in different areas of the graph was further investigated by plotting RCPs per individual cells for cells located in different sections of the graph (Supplemental Fig. 4). The data show that cells in all areas of the graph, corresponding to all phases of the cell cycle, respond to stimulation and again reflect the observations for interphase cells and mitotic cells described above. In summary, the scatter plots show that large G1/S/G2 phase cells comprise most of the cells with the highest levels of linker-phosphorylated Smad2 and also that a selection of mitotic cells (4 N cells with small nuclei) possess increased phosphorylation levels (Fig. 2c, d (0 h and 1 h) and Supplemental Fig. 2–4). These observations are in line with earlier studies of the related T179 linker phospho-site of Smad3[21], and with the formation and localization of specific Smad complexes in mitotic cells[27,28].

We also analyzed the effects of proteasomal inhibition by MG132 in individual cells according to the size and DNA content of their nuclei, plotting HaCAT cells stimulated with TGF-β in the presence or absence of MG132 (Fig. 3a–d). MG132 resulted in a robust increase of all linker-phosphorylated isPLA products after 6 h of treatment. Large G1/S/G2-cells and small 4 N mitotic cells maintained the highest phosphorylation levels also after proteasomal inhibition (Fig. 3a–d).

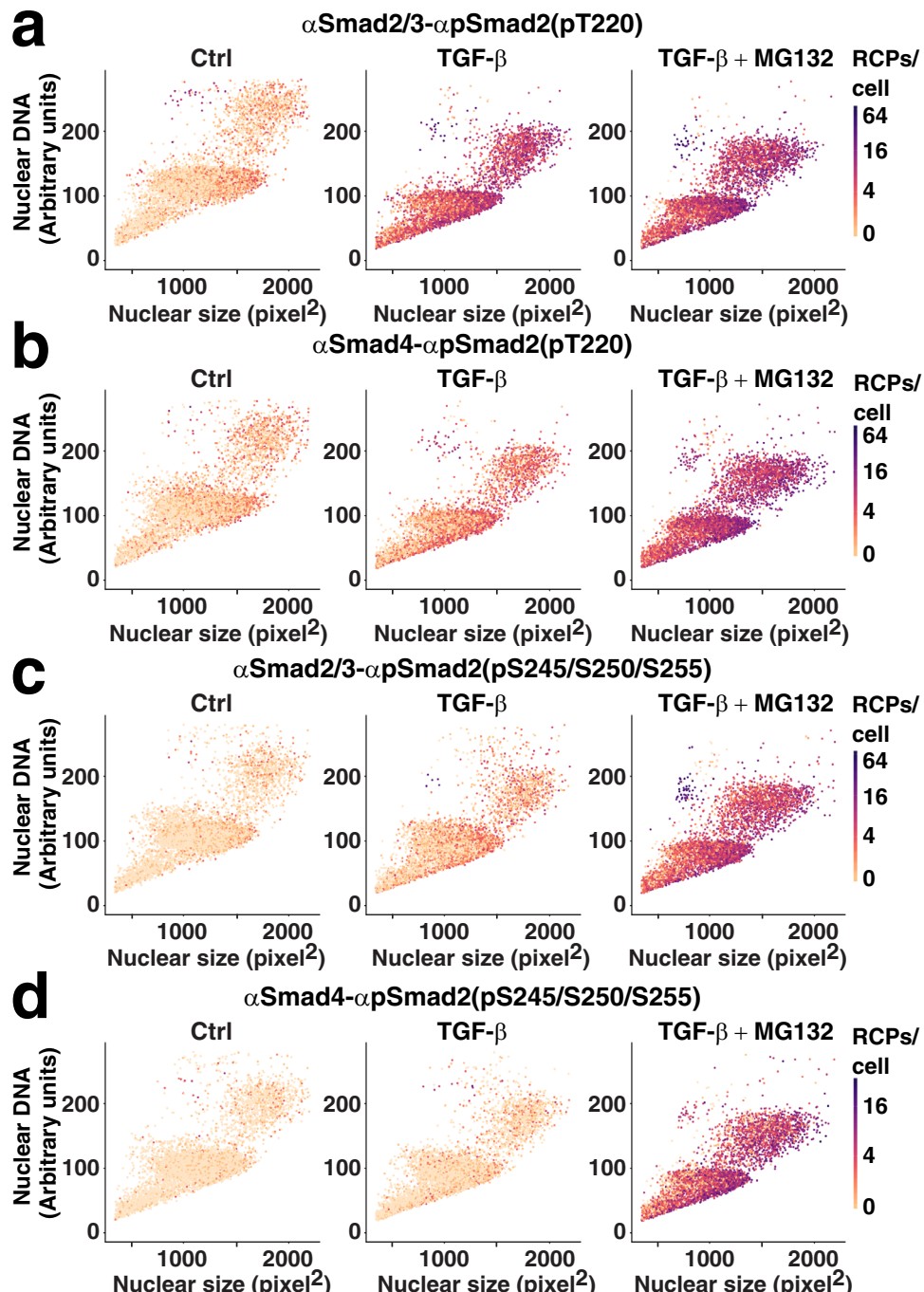

**Fig. 3 Smad2 linker phosphorylation and complex formation in response to TGF-β and the proteasomal inhibitor MG132.** HaCAT cells were starved and untreated, treated with 10 ng/ml TGF-β for 6 h, or treated with 10 ng/ml TGF-β and 25 μM MG132 for 6 h before fixation, isPLA, and imaging. Individual cells were color-coded according to the numbers of RCPs/cell for **a** Smad2/3-Smad2(pT220), **b** Smad4-Smad2(pT220), **c** Smad2/3-Smad2(pS245/pS250/pS255), and **d** Smad4-Smad2(pS245/pS250/pS255). The cells were plotted according to the area of the nuclear stain along the *X* axes (pixels$^2$ (~0.11 μm$^2$)) and the integral intensity of the DNA stain along the *Y* axes. The log2 color scales indicate 0–64 RCPs/cell for **a–c** and 0–32 RCP/cell for **d**.

We further investigated differences in signaling intensity of cells in relation to their crowding/density after 1 h of TGF-β stimulation as reflected in Smad2(pT220) levels. For this purpose, we plotted cells along the *X* axis according to the estimated percentage of the circumference of each cell that was in contact with other cells (calculated based on nuclear position and the average size of HaCAT cells). The cells were plotted along the *Y*

axis according to their estimated numbers of neighboring cells, as inferred via the numbers of nuclei present within a defined distance of another nucleus. We selected a distance of 20 pixels (~6.6 μm), which provided a distribution of nearby nuclei generally ranging from 0 to 5 neighbors. The data show that the fewer close neighbors a cell has, the higher its signaling activity (Fig. 4a). This observation becomes more prominent

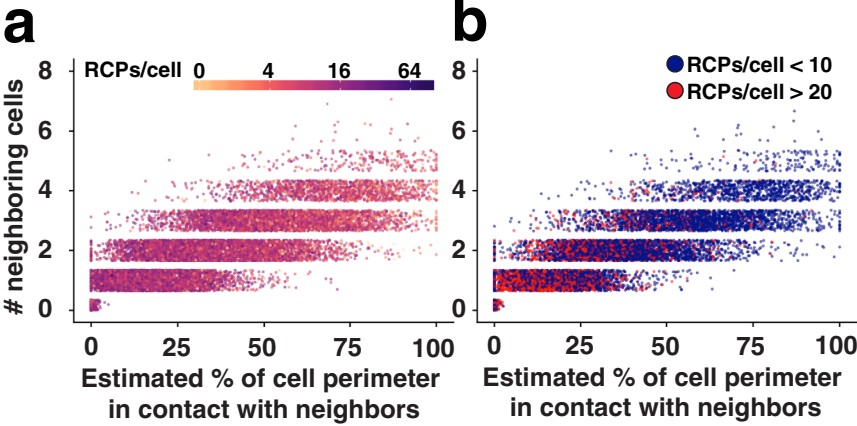

**Fig. 4 Smad2 pT220 linker phosphorylation levels in relation to cell crowding/density after 1 h of TGF-β stimulation.** Graphs display the estimated percentage of the circumference of each cell that is in contact with other cells (calculated based on nuclear positions of cells and the average size of HaCAT cells) along the X axis and estimated numbers of neighboring cells as inferred via the numbers of nuclei present within a defined distance (20 pixels (~6.6 μm)) along the Y axis. Graph **a** displays all cells while graph **b**, for ease of view and clarity, only displays cells with RCPs >20 as red symbols and RCPs <10 as dark blue symbols.

when the graph only displays the cells with the highest or lowest intensity of signaling (RCPs> 20; red symbols vs. RCPs <10; dark blue symbols) after 1 h of TGF-β stimulation (Fig. 4b). The notion that TGF-β signaling can be affected by local cell crowding has been studied and reported by others[27,29,30].

In summary, by combining isPLA with high-throughput image analysis of individual cells, we monitored Smad phosphorylations and interactions throughout the cell cycle and in relation to their proximity to neighboring cells. The analysis allowed us to define distinct characteristics of strongly or weakly responding individual cells.

**Screening effects of phosphatase inhibitors by semi-automated high-throughput isPLA.** The high-throughput analysis of specific cell-signaling events via isPLA provides an opportunity to screen libraries of drug candidates and other agents for their effects on specific molecular processes, distinguishing effects on individual cells. As a proof of principle, we investigated how a panel of 33 experimental phosphatase inhibitors would influence the levels of Smad2(pS245/pS250/pS255). We observed that three of the investigated compounds, NSC-95397, Shikonin, and NSC-663284, increased the levels of Smad2(pS245/pS250/pS255) even without TGF-β stimulation (Fig. 5a). Interestingly, these three inhibitors are all quinone-type compounds that have previously been shown to target Cdc25[31–33]. We confirmed the upregulation of Smad2(pS245/pS250/pS255) by western blot after the addition of NSC-663284 alone, observing an increased phosphorylation level in bands of the correct molecular weight (Fig. 5b). Furthermore, we observed that this increase could be partially blocked by the TGF-β receptor inhibitor GW6604 and completely blocked by the MEK inhibitor U0126, whereas it was not affected by inhibitors of the GSK3 (SB-216763) or Cdk (flavopiridol) pathways (Fig. 5b). We conclude that quinone-type compounds, like NSC-663284, lead to higher phosphorylation levels by endogenous MEK and/or TGF-β signaling. This may suggest that Cdc25 could play either a direct or indirect (via control of other signaling components of the MEK or TGF-β pathways) role in regulating Smad2 linker phosphorylations. Further studies are needed to investigate the specificity, mechanism, significance, and potential applications of these quinone-type inhibitors on the regulation of linker-phosphorylated Smad2.

## Discussion

We present a high-throughput approach for a high-content assessment of cell signaling. We show that the method can be used in a 96-well format to screen signaling dynamics across millions of individual cells in response to cytokine stimuli or inhibition by members of drug libraries. The technique compares very favorably to the time-consuming work of preparing individual tests as needed for most other endogenous interaction assays. Compared with flow cytometry and CyTOF, this approach allows for a combined evaluation of cellular features that are normally lost owing to trypsinization of cells, including cell shape, size, and contacts, as well as parameters such as local cell crowding. The protocol also permits individual cells to be revisited after data analysis for further molecular or morphological analyses. In summary, the technique provides an avenue for detailed characterization of molecular events and cellular processes, producing precise spatial and temporal signaling maps.

The isPLA technique does not depend on transfections or genetic engineering of cells, rendering the method applicable for studying proteins, modifications, and interactions in cell lines, primary cells, or diagnostic patient materials alike. The fact that the assay requires fixation confers both advantages and some limitations. On one hand, the fixed cells can serve as a permanent archive that can be revisited for reanalysis or even further assayed with new probes. Diagnostic materials, such as tissue sections are typically stored after fixation and can be investigated with the assay protocol described herein. An obvious limitation is that the analysis of fixed materials by isPLA, in contrast to interaction assays such as BiFC, BioID, APEX, or FRET, can only provide snapshots, unsuitable for tracking live cells[2]. Assays used to monitor interactions in live cells, on the other hand, often have their own limitations, such as e.g. the need for transfections or genetic engineering.

Here, we applied a semi-automated isPLA approach to explore how linker-phosphorylated Smad2 is regulated. We charted precise time courses of cell stimulation as reflected in levels of Smad2(pT220) and Smad2(pS245/pS250/pS255) after TGF-β stimulation and recorded interactions with Smad4 (Fig. 1 and supplemental Fig. 1). The responses to stimulation and drug treatment were characterized at the single-cell level, and related to several morphological parameters (Figs. 2–5). We found that G1/S/G2 phase cells with large nuclei showed the highest

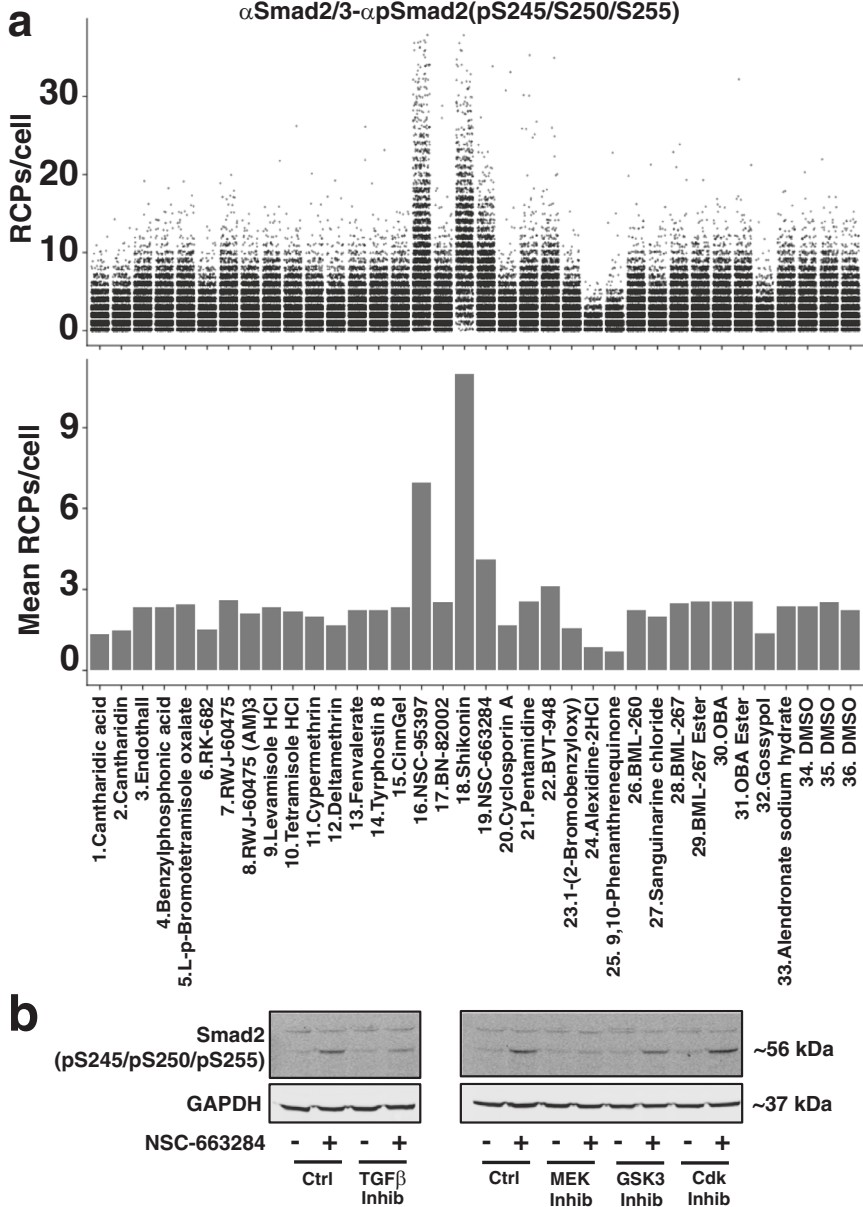

**Fig. 5 Effects by members of a library of 33 phosphatase inhibitors on Smad2/3-Smad2(pS245/pS250/pS255) levels in HaCAT cells. a** HaCAT cells were treated or not with 10 µM of the indicated phosphatase inhibitors for 4 h before fixation, isPLA, and imaging. Displayed as individual cells plotted according to their numbers of Smad2(pS245/pS250/pS255) RCPs/cell (top), or as bar graphs showing mean RCPs/cell for each sample. **b** Western blots for HaCAT cells treated with or without the inhibitor NSC-663284 and with or without the specific pathways inhibitors (TGF-β inhibitor (GW6604), MEK inhibitor (U0126), GSK3 inhibitor (SB-216763), or Cdk inhibitor (flavopiridol)) to block potential endogenous signal transduction. The blots display levels of Smad2(pS245/pS250/pS255) (top panel) and the housekeeping gene GAPDH as control (bottom panel). Full western blot scans are included in supplemental Fig. 5.

phosphorylation signals in response to TGF-β stimulation. We also demonstrate that cells entering mitosis, with high DNA content corresponding to 4 N and nuclear envelopes that break down, displayed increased phosphorylation of the investigated proteins, which is in line with earlier reports[21,27,28].

In conclusion, the high-throughput approach allowed us to visualize Smad2 phosphorylation throughout the cell cycle, and to characterize responses by the cells in relation to a range of cell biological parameters through automated image analysis. Finally, we also illustrated the possibility to use this high-throughput analysis to screen for drug effects on defined molecular events, in a manner that will be of particular value for evaluating heterogeneous cell populations such as primary patient samples. Accordingly, this high-

throughput isPLA protocol enables in situ analyses of protein complexes and PTMs under large numbers of conditions with limited hands-on work and with the possibility to revisit the permanent record of specific cells of interest for further in situ analyses.

## Materials and methods

**Antibodies and inhibitors**. Anti-GAPDH (AM4300) (WB 1:30000) was purchased from Life Technologies. Anti-Smad2(pT220) (sc-135644) (isPLA 1:100) 1: and anti-Smad4 (sc-7966) (isPLA 1:100) were from Santa Cruz Biotechnologies. Anti-Smad2(pS245/pS250/pS255) (#3104) (WB 1:1000, isPLA 1:150) was from Cell Signaling and anti-Smad2/3 (610842) (isPLA 1:150) from BD Bioscience. The SCREEN-WELL phosphatase inhibitory library (BML-2834.0100) was from Enzo Life Sciences. MG132 (sc-201270), GSK3 inhibitor SB-216763 (sc-200646), and flavopiridol (sc-202157) were purchased from Santa Cruz Biotechnologies, and

MEK inhibitor U0126 (#9903) was acquired from Cell Signaling. The TGF-β receptor inhibitor was a kind gift from the Ludwig Institute for Cancer Research and has been described before[21].

**Cell culture, cell stimulation, fixation, and permeabilization.** HaCAT cells were maintained in Dulbecco's Modified Eagle Medium (Gibco) supplemented with 10% fetal bovine serum (Gibco), 100 U/ml penicillin (Gibco), and 100 U/ml streptomycin (Gibco). For experiments, cells were seeded into half-area μclear 96-well plates from Greiner (675986). Cells were serum-starved before stimulation with 5 or 10 ng/ml TGF-β with or without inhibitors up to the indicated time points. MG132 was used at 25 μM and the phosphatase library was used at 10 μM concentration. After stimulations, cells were washed twice in phosphate-buffered saline (PBS; pH 7.4), fixed for 10 or 12 min with 3% paraformaldehyde in PBS on ice, washed three times with PBS, and permeabilized for 10 min with 0.4% or 0.5% Triton x-100 in PBS at room temperature on a shaker.

**isPLA and imaging.** For isPLA we used Duolink anti-rabbit minus (DUO92005), Duolink anti-mouse plus (DUO92001) and Duolink detection reagents red or green (DU92008) purchased from Olink Bioscience and Sigma Aldrich. In brief, the fixed and permeabilized cells were blocked with Duolink blocking buffer for 2 h at 37 °C or overnight at 4 °C. Primary antibodies were incubated overnight in Duolink antibody diluent and subsequently washed intensively with Tris-buffered saline, pH 7.4, and 0.05% (vol/vol) tween-20 (TBS-T). Duolink secondary antibodies were incubated on cells for 2 h at 37 °C and subsequently washed intensively with TBS-T (0.05%). Ligation, rolling-circle amplification, and subsequent washes were performed according to the Duolink standard protocol. Cells were then stained with Hoechst 33342 before being assayed on a Molecular Devices Imagexpress-micro XL automated microscope at ×20 magnification (Nikon S Plan Fluor 20 × 0.45NA) and MetaXpress software. The pixel size for images acquired at ×20 magnification is 0.33 × 0.33 μm (Molecular Devices user guide https://www.moleculardevices.com/sites/default/files/en/assets/user-guide/dd/img/imagexpress-micro-4-widefield-high-content-imaging-system.pdf). For each well, images were acquired from nine fields of view. For the displayed images, contrast adjustments have been performed for optimal visualization with the same exact treatment of all images within each displayed dataset.

**Cell profiler.** Image analysis was performed with the CellProfiler version 3 software package. For optimal identification and enumeration of individual nuclei and RCPs, manual size gates and intensity thresholds were validated and applied according to the pipeline described below. The sizes of HaCAT nuclei and isPLA RCPs are consistent between experiments and therefore enable the use of a robust standard detection protocol for these parameters by setting up the background reducing tool of the CellProfiler program (EnhanceOrSuppressFeatures) in combination with defined standard size limits (see pipeline below). The intensity thresholds, on the other hand, are affected by multiple experimental parameters and therefore need to be individually defined for each experiment (i.e., for each individual microtiter plate). To set intensity thresholds, an image in the first quadrant of the microtiter plate is selected and the general intensity range of the nuclear Hoechst stain (background vs nuclei) and isPLA fluorescence stain (background vs RCP) is inspected. A manual threshold is then set (above the background range but below the lowest identified true nuclei/RCP intensity). Before initiating the automated microscope run, the size and intensity thresholds are validated by inspecting a graphically visualized CellProfiler image from each quadrant of the plate. Because of the controlled plate density of the HaCAT cells and their normal growth with relatively discrete nuclei, along with the generally limited numbers of RCPs per cell for cellular signaling events, there is only a small risk that nuclei and RCP counts represent underestimates owing to e.g., overlapping signals.

The pipeline included the following modules and key parameters: EnhanceOrSuppressFeatures (Speckles size 50 (nuclei) (~16.5 μm), IdentifyPrimaryObjects (nuclei, manually defined intensity threshold, 20–60 pixel size limit (~6.6–19.8 μm)), IdentifySecondaryObjects (cytoplasm, see information below), EnhanceOrSuppress Features (Speckles size 5 (RCPs) (~1.7 μm), IdentifySecondaryObjects (RCPs, manually defined intensity threshold, 1–10 pixel size limit (~0.33–3.3 μm)) Mask Objects (cells or nuclei), MeasureObjectSizeShape (nuclei), MeasureObjectNeighbors (2 × 10 pixels (~6.6 μm)), RelateObjects and ExportToSpreadsheet. To evaluate the cytoplasmic location of RCPs, an area within 60 pixels (~19.8 μm) surrounding each nucleus was assigned as cytoplasm. This distance was decided based on prior collected observations from phalloidin staining and phase-contrast images of the sizes of HaCAT cells. If two nuclei are located within 120 (2 × 60) pixels (~40 μm) or less, then the border between them is drawn in the middle, and no RCPs are counted twice. In fact, HaCAT cells grow in dense clusters and the distance between nuclei is in most cases substantially shorter. This is illustrated by the fact that when measuring neighboring cells in Fig. 4, the distance that was used to get a good distribution of neighboring nuclei was only 20 (2 × 10) pixels (~6.6 μm). Data were collected with the ExportToSpreadsheet function. Visualizations of data were done using the R-Studio software by allowing the software to plot individual cells along the X and Y axes and color-coding them according to the information stated in each figure.

**Western blot.** Cells in culture dishes were stimulated and inhibited as indicated, and lysates were prepared by gently washing the cells in ice-cold PBS before adding 1.5× NuPAGE LDS Sample Buffer with dithiothreitol (100 mM). The samples were then scraped, collected, sonicated, and heat-denatured at 95 °C for five min. The lysates were run on SDS-PAGE and the resulting gels were soaked for 10 min in 50 mM Tris base, 40 mM glycine, 0.037% SDS, and 20% methanol before being transferred to polyvinylidene difluoride membranes using an IBlot2 device. Transferred membranes were blocked for two hours in 1:3 diluted Odyssey block (LI-COR Biosciences) in TBS and incubated overnight with anti-Smad2(pS245/pS250/pS255) (#3104, cell signaling) antibodies at 4 °C. Next, the membranes were washed in TBS-T, incubated with IRdye-labeled secondary antibodies (LI-COR Biosciences), and then washed again with TBS-T, with a final wash in TBS. Blots were scanned and visualized with the Odyssey Scanner and ImageStudio Lite system (LI-COR Biosciences).

**Statistical analysis and reproducibility.** Quantitative data are, where possible, displayed as the mean ±SD according to the details laid out in the respective figure legends.

**Reporting summary.** Further information on research design is available in the Nature Research Reporting Summary linked to this article.

## Data availability

The source data for graphs and charts of key figures in the main article are available as Supplementary Data 1. Further information can be obtained from the corresponding author upon reasonable request.

## Code availability

R studio (version 1.1.423) and MacBook Pro OS X (version 10.9.5) have been used to visualize data by allowing the software to plot individual single-cell data points along the X and Y axes and color-coding them according to the information stated in each figure. Code is available from the corresponding authors upon reasonable request.

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

## Acknowledgements

We thank A. Zieba, O. Söderberg, A. Blokzijl, C. Gallant, and A. Klaesson for valuable input. This work was supported by grants to UL from the Swedish Research Council (2012-5852, 2017-04152, 2018-02943, and 2018-06156), IngaBritt och Arne Lundbergs Forskningsstiftelse, The Swedish Foundation for Strategic Research (SB16-0046), Torsten Söderbergs Stiftelse (M130/16), The Swedish Collegium for Advanced Studies (SCAS), The Swedish Cancer Society (19 0384 Pj) and the European Research Council under the European Union's Seventh Framework Programme (FP/2007-2013)/ERC Grant agreement no. 294409.

## Author contributions

P.L. conceived, designed, and performed experiments, and analyzed data; R.A.A.-A., E.M.D., J.H. performed experiments and analyzed data; R.G., J.B., J.O., M.K-M. analyzed data; U.L. conceived of the concept and analyzed data; P.L., R.A.A.-A. and U.L. wrote and edited the article. All authors reviewed and commented on the manuscript.

## Funding

## Competing interests

Ulf Landegren is a co-founder and shareholder of Navinci, having rights to the isPLA technology. All other authors declare no competing interests.
