## [Transparent Peer Review File · Communications Biology]

Reviewers' comments:

Reviewer #1 (Remarks to the Author):

This article proposes an interesting method for single cell study of cell signaling on large cell populations based on direct microscopy imaging. In situ proximity ligation assay was combined with high throughput imaging to determine Smad linkers phosphorylation and their interaction, in particular under the effect of TGFbeta.

The paper serves essentially a methodological purpose as several biological insights have been already exposed in the literature, as cited in the manuscript. The method provides however potentially more detailed observations. The drug screening application is interesting but the authors don't specify if another method like flow cytometry would have been applicable here. While the description of the method may be intended to be practically usable by non-experts, it does not prevent the authors from:

- describing more precisely certain aspects of the protocols
- rendering the results in a more readable or rigorous form
- performing adequate controls

Specific comments:

1. for reproducibility physical distances should not only be expressed in pixels but also in μm .
2. the segmenting of the RCP should be better described and the choice of threshold distances discussed.
3. the scatter plots of fig 2, 3 representing nuclear stain vs nuclear size exhibit regular boundaries which possibly reflect arbitrary choices of size/intensity threshold in the data analysis. The potential influence of those choices and artifacts on the results should be discussed.
4. The relation between nucleus measurements and the cell cycle should be better justified. For example, in the reference 26, nucleus staining was performed with DAPI, while Hoechst labelling is used here. A control of cell cycle with a fluorescent indicator would be welcome.
5. Most conclusions from fig. 2., 3, 4 are drawn from graphical representations using a color code for RCP. Please quantify and test the statistical difference also in a conventional way in order to present a synthetic biological outcome. For example, tests within gated areas of the plots could be performed.
6. The information contained in the representation of violin plot with a color code indicating superposing RCP is not clear. An alternative representation of fluorescent signal of isPLA in terms of intensity could also be exploited and maybe more robust.

Reviewer #2 (Remarks to the Author):

Comments:

In this article, the authors establish an extremely sensitive and specific, high-throughput semi-automated platform for interrogating the dynamics protein interactions during cell signalling in a pseudo spatio-temporal manner. The method combines the power of the well-established in situ proximity ligand assays with automated microscopy and computer-based image analysis to then derive spatio-temporal post-translational modifications and complex formation information from over a million individual cells at single cell resolution. They apply the methodology to study the phosphorylations and complex formations of Smad2 linker regulated by TGF-beta and various drugs on cultured HaCAT keratinocyte cells. This work presents a significant advance in the field of single-cell characterization by the methodology's ability to provide additional spatial information such as cell morphology, location of cells and their local microenvironment that is lost in a FACS experiment. The results clearly meet the requirements and the goals set in the work and the manuscript will attract significant attention from the field in future.

Remarks:

There are a few recommendations and concerns that can be addressed to improve the manuscript in its current state and warrant its publication in Communications Biology:

1. Is there a reliable way to estimate nuclear volume from the 2D images? Does the shape of the nuclei play a role? How do the authors separately count the RCP numbers inside the nucleus and in cytoplasm from the 2D images? Can the authors comment on the cell morphology during the various stages in cell cycle?
2. At high RCP densities in the 2D images, the emission from neighbouring RCPs seem to overlap significantly. Can the authors comment on how a reasonable estimate of the numbers are achieved in such cases?
3. Could the authors include the units in the axis labels for Nuclear size? Do the numbers indicate area in terms of pixels? Marking (roughly) the areas corresponding to the various phases of cell cycle in one exemplary Nucl. Size – Nucl. Stain graph could serve the readers in providing a better visualization.
4. With the huge amount of statistics acquired in this work, can the authors further comment if the local cell crowding effects is cell cycle phase dependent? Do the cell cycle phase of the neighbouring cells effect the TGF-beta signalling?
5. It would be very interesting if the authors can identify if there are any patterns in the spatial distribution of the Smad complexes within the nucleus as a function of cell cycle phase.
6. One particular limitation of in situ proximity ligand assays is that it requires cell fixing that does not allow monitoring the same cells in time. May I request the authors to discuss this in the conclusions and maybe suggest a few future alternatives?

We thank the Reviewers for their insightful and helpful comments and questions. Please find our responses to their questions below. Updated version of the manuscript (**COMMSBIO-21-0826**) where the changes have been highlighted is appended to this resubmission.

Reviewer #1 (Remarks to the Author):

Specific comments:

1. for reproducibility physical distances should not only be expressed in pixels but also in μm .

= We agree, and the relationship between pixels and μm has now been clarified in the manuscript, materials and methods, figures and figure legends in relation to distances and nuclear sizes on page 5 (lines 9-11), page 7 (lines 20-22, 43-46), page 8 (lines 1-10), page 9 (line 37), page 10 (lines 4, 12, 44) and page 11 (line 12). Moreover, we have now added scale-bars to the micrographs in figure 1B, 1E, Suppl fig 1C and Suppl Fig 3.

2. The segmenting of the RCP should be better described and the choice of threshold distances discussed.

= The procedure for manually setting and confirming the thresholds for RCPs and nuclei identification, including the specific settings in CellProfiler, is now further clarified in the Materials and Methods section on page 7 (lines 26-46) and page 8 (lines 1-13).

3. The scatter plots of fig 2, 3 representing nuclear stain vs nuclear size exhibit regular boundaries which possibly reflect arbitrary choices of size/intensity threshold in the data analysis. The potential influence of those choices and artifacts on the results should be discussed.

= The procedure for manually setting and confirming the thresholds for RCPs and identifying nuclei, including the specific settings in CellProfiler, is now described in the Materials and Methods section on page 7 (lines 26-46) and page 8 (lines 1-13). Furthermore, the boundaries in the scatter plots of Figs. 2B, 2C, 3, and Suppl. fig. 2 are also discussed in the manuscript on page 4 (lines 5-20 and 28-41).

4. The relation between nucleus measurements and the cell cycle should be better justified. For example, in the reference 26, nucleus staining was performed with DAPI, while Hoechst labelling is used here. A control of cell cycle with a fluorescent indicator would be welcome.

= To further demonstrate the relation between the graphs and the cell cycle, representative images of cells in different phases of the cell cycle are now included together with their positions in the plotted graph. These data are included as Supplemental Figure 3 and discussed on page 4 (lines 28-34) and in new figure legends on page 11 (lines 4-8). These images confirm where mitotic cells, newly divided cells, and larger nuclei are primarily located in the graph. In combination, this serves as a confirmation of the inferred cell cycle stages in the graphs. The use of DAPI versus Hoechst is also discussed.

5. Most conclusions from fig. 2., 3, 4 are drawn from graphical representations using a color code for RCP. Please quantify and test the statistical difference also in a conventional way in order to present a synthetic biological outcome. For example, tests within gated areas of the plots could be performed.

= A new figure with graphs that compare the TGF- β signaling between gated cells located in different areas of the plots is now included as Supplemental Figure 4, referred to on page 4 (lines 34-38) and in new figure legends on page 11 (lines 10-18), to further support and add to the discussions in the paper.

6. The information contained in the representation of violin plot with a color code indicating superposing RCP is not clear. An alternative representation of fluorescent signal of isPLA in terms of intensity could also be exploited and maybe more robust.

= We agree that the purpose of the graphs in the submitted version was not sufficiently clear and they have now been removed. The superimposed violin graphs were in fact inaccurate because of the overlapping data points that were shown in different colors. The earlier dot plot graphs have therefore now been replaced by bar graphs in Figure 2A and 2B, referred to on page 3 (lines 40-46) and in the figure legends on page 9 (lines 32-35), to more clearly illustrate the variation of numbers of RCPs among individual unstimulated and stimulated cells.

Reviewer #2 (Remarks to the Author):

Remarks:

There are a few recommendations and concerns that can be addressed to improve the manuscript in its current state and warrant its publication in Communications Biology:

1. Is there a reliable way to estimate nuclear volume from the 2D images? Does the shape of the nuclei play a role? How do the authors separately count the RCP numbers inside the nucleus and in cytoplasm from the 2D images? Can the authors comment on the cell morphology during the various stages in cell cycle?

= These are valid points. We now clarify on page 3 (lines 12-15) that the determination of nuclear RCPs is based on a 2D projection and may also include some cytoplasmic signals located above or below the nuclei. The 2D projection also precludes calculating the nuclear volume. Regarding the shape of the nuclei, we now include a Supplementary Figure 3 representative images of the nuclei with their RCPs in various stages in the cell cycle and we include a short description in the manuscript on page 4 (lines 28-34) and in new figure legends on page 11 (lines 4-8).

2. At high RCP densities in the 2D images, the emission from neighbouring RCPs seem to overlap significantly. Can the authors comment on how a reasonable estimate of the numbers are achieved in such cases?

= It is correct that when RCPs are dense the software may fail to distinguish individual RCPs. The procedure for manually setting and confirming the threshold for RCP identification in the CellProfiler software and avoiding overlaps is now described in the material and methods section on page 7 (lines 26-46) and page 8 (lines 1-13). We also described there that because of the generally limited numbers of RCPs per cells for the investigated cellular signaling events, there is only a small risk that RCP counts represent significant underestimates of the true numbers of RCPs.

3. Could the authors include the units in the axis labels for Nuclear size? Do the numbers indicate area in terms of pixels? Marking (roughly) the areas corresponding to the various phases of cell cycle in one exemplary Nucl. Size – Nucl. Stain graph could serve the readers in providing a better visualization.

= We agree with this comment and have now updated the manuscript accordingly. The relationship between pixels and μm has been clarified in the manuscript, materials and methods, figures and figure legends in relation to distances and nuclear sizes. Updates are introduced on page 5 (lines 9-11), page 7 (lines 20-22, 43-46), page 8 (lines 1-10), page 9 (line 37), page 10 (lines 4, 12, 44) and page 11 (line 12). Moreover, we have now added scale-bars to the micrographs in figure 1B, 1E, Suppl fig 1C and Suppl Fig 3.

To further demonstrate the relation between the position of individual cells in the cell cycle and the graphs reporting nuclear size versus DNA staining we now include representative images of cells in different phases of the cell cycle along with the location of those cells in the plotted graph. These data are included as a Supplemental Figure 3 discussed on page 4 (lines 28-34) and in new figure legends on page 11 (lines 4-8). The images help illustrate where mitotic cells, newly divided cells, and larger nuclei are located in the graph. The images thus serve to confirm the inferred cell cycle stages in the graphs.

4. With the huge amount of statistics acquired in this work, can the authors further comment if the local cell crowding effects is cell cycle phase dependent? Do the cell cycle phase of the neighbouring cells effect the TGF-beta signalling?

= We thank the reviewer for this interesting question regarding cell cycle phases of the neighboring cells and their effect on each other. However, at this point, the assay and data generation can only reveal cell cycle phases at the population level. It would however be an interesting topic for further analyses.

5. It would be very interesting if the authors can identify if there are any patterns in the spatial distribution of the Smad complexes within the nucleus as a function of cell cycle phase.

= To further demonstrate the relationship between the graphs and the cell cycle, representative images are now included as a Supplemental Figure 3 discussed on page 4 (lines 28-34) and in new figure legends on page 11 (lines 4-8). The images display the nuclei and RCPs in various stages in the cell cycle are also discussed in the manuscript page 4 (lines 28-34).

6. One particular limitation of in situ proximity ligand assays is that it requires cell fixing that does not allow monitoring the same cells in time. May I request the authors to discuss this in the conclusions and maybe suggest a few future alternatives?

= We agree with the reviewer and have now underlined some advantages and disadvantages of analyses of fixed cells, and we highlight additional relevant techniques in the Discussion section on page 6 lines 8-17.

REVIEWERS' COMMENTS:

Reviewer #1 (Remarks to the Author):

The authors have satisfactorily addressed my comments.

Reviewer #2 (Remarks to the Author):

The manuscript, together with the modifications addressing the comments raised during revision process, is highly convincing and will contribute significantly to high throughput image-based protein interactions and cell morphology studies in the future. In my opinion it is written in a very clear manner with adequate details to reproduce the work and will attract significant attention in the field.